# Structural Variations in the Genome of Potato Varieties of the Ural Selection

**Georgiy A. Lihodeevskiy ***  **and Elena P. Shanina**

Ural Federal Agrarian Research Center, Ural Branch of the Russian Academy of Science,
620142 Ekaterinburg, Russia; shanina08@yandex.ru
* Correspondence: georglihodey@gmail.com

**Abstract:** Potato (*Solanum tuberosum* L.) is the third most common plant crop in the world. Many studies, such as those using marker-assisted selection (MAS), are devoted to the genomic evaluation of potato. However, the nucleotide composition of some markers has not been described yet, and some regions of the genome remain unknown. Thus, the development of new marker systems for further genetic selection is required. Whole-genome sequencing and the search for structural variants (SVs) should further develop molecular genetic studies of the potato crop. In this paper, we will show the first results of our studies of the three backcrossed varieties Alaska, Argo, and Shah, which were selected for sequencing. Alaska is a patented variety with confirmed resistance to pathogens, while Argo and Shah are new perspective varieties. We sequenced genomes of these varieties using a nanopore platform. As a result, we identified more than 24,000 authentic structural variants with lengths varying from 4 bp to 100 Mbp. Deletion was found to be the most common type of structural variant in the genome and the genes. The majority of SVs are located in non-coding regions, including introns. However, a quarter of the genes of the sequenced varieties have some chromosomal mutations. Some genes responsible for resistance to abiotic stress and pathogens were duplicated, while genes of nucleic acid polymerization and few metabolic proteins were deleted.

**Keywords:** potato *Solanum tuberosum* L.; structural variants; potato genome





## 1. Introduction

Potatoes have a long history of cultivation and improvement around the world. The main selection lines in potato breeding are focused on resistance to pathogens (viruses, fungi, and bacteria), protection against abrupt climatic changes, and improvement of nutritional quality [1]. Many wild potato species can be crossed with cultivated species and varieties, creating a rich genetic breeding pool [2]. Approximately 40% of wild species are carriers of genetic traits that defend against pests and diseases [3]. However, modern varieties carry only a fraction of the beneficial genes of their wild relatives. Among these genes, the most significant are related to resistance to viruses (*Solanum stloniferum*, *S. tuberosum* ssp. *andigena*), nematodes (*S. spegazzini, S. vernei*), and blight (*S. demissum, S. bulbocastanum*) [4–7]. For a long time, recurrent selection by phenotype remained the only method of potato breeding [8]. This type of breeding is time-consuming and takes up to 30 years to obtain a variety [6]. Its main limiting factors include the quantitative nature of essential traits, inbreeding depression, low selection intensity in the early generations, low pollen fertility of cultivated varieties, and the accumulation of pathogens in hybrids [1,9].

Marker-assisted selection (MAS) allows early selection of hybrids to identify essential genes and loci of quantitative traits (QTLs) and is widely used to increase the rate of selection [10,11]. A wide range of genetic markers have been developed, for example, to determine resistance to X and Y viruses [12–15], nematodes [16,17], *Phytophthora* [18,19], and potato wart (*Synchytrium endobioticum*) [20]. Most of these markers were created using two main techniques: amplified fragment length polymorphism (AFLP) or restriction fragment length polymorphism (RFLP), i.e., without known sequences. All of them are

related to the presence or absence of the PCR product, which means they are connected to structural variants, such as an insertion or a deletion.

Before deciphering the potato genome, various genetic markers were used "blindly" by breeders, while the association of candidate genes underlying the traits (although analyzed with QTL) remained unknown. The potato genome was first sequenced in 2011 by the shotgun method using the Illumina and Roche platforms, amended with Sanger sequencing data. Due to potato species' polyploidy (two to five sets of chromosomes), the genome of the doubled monoploid clone DM 1-3 516 R44 (hereafter DM) was sequenced to facilitate genome assembly. As a result, 86% of the 844 Mb genome was covered. The assembly included 12 pseudochromosomes with 39,031 predicted protein-coding genes [21]. Three reference assemblies were published during the potato genome sequencing research. In 2011, DM assembly v4.03, comprised of 12 pseudochromosomes, was aligned and compared to the available tomato genome. Later, DM v4.04 included new contigs from whole-genome sequencing and augmented with unaligned reads. DM v6.1 was improved by nanopore sequencing of the doubled monoploid genome and application of Hi-C technology These updates made it possible to obtain longer contigs, reduce gaps in the nucleotide sequences, and, as a result, improve sequence completeness [22]. The existence of a reference genome is vital for further selective breeding programs. There is another challenge in the study of the genome and the search for associations. Potato is a clonally propagated plant, and it carries significant variability in the number of structural variant (SV) copies and deletions or duplications in one-third of its genes [23]. Sequencing and searching for SVs allow a detailed description of the potato genome to be made and predict adverse mutations [24]. Moreover, this can help to understand some of the adaptations. There is evidence that cultivated potato has increased copy numbers of disease resistance and abiotic stress genes. However, most SVs lie outside of the coding sequences [25]. In addition, the study of SVs helps create genetic markers in flanking non-coding parts of the target to increase specificity [26].

In the Ural region, preference is given to varieties with a short vegetation period due to climatic conditions. Further, widespread potato wart and late blight are the most common potato pathogens in the Urals. The study of pathogen resistance in the Sverdlovsk region primarily attracts our attention. Thus, new hybrids and varieties are tested for resistance to potato wart, nematodes, *Phytophthora sp.*, and X and Y viruses. The Alaska variety, patented in 2020, is resistant to potato wart, *Globodera rostochiensis* [27], and to late blight in tubers and leaves. Alaska has elongated tubers with red skin and white flesh which becomes crumbly after cooking; the starch content is 14.0–18.5%; the yield can be harvested until October. Argo and Shah are new varieties also resistant to potato wart. Argo has red skin, white flesh and a rounded tuber; Shah has the same tuber shape but yellow skin and flesh. All the varieties have a complex of economically valuable traits (resistance to biotic and drought stress, starch content, etc.), making them suitable crops in the Urals and similar regions.

## 2. Materials and Methods

### 2.1. Plant Materials

Our study focuses on three varieties of potatoes of the Ural selection—Alaska, Argo, and Shah. All of the varieties are from the backcrosses line and have some common ancestors. For DNA extraction, we used young tetraploid potato plants grown in a sterile agar medium.

### 2.2. Genomic DNA Isolation and Purification

DNA was extracted from the plants with the innuPREP Plant DNA Extraction Kit by Analytik Jena (Jena, Germany) according to protocol #3. Before isolation, samples were homogenized in tubes with zirconium beads. Before sequencing, the resulting DNA eluate was cleaned of residual RNA using the following procedure:

1. 5 μL of RNase Cocktail™ Enzyme Mix by Invitrogen (Carlsbad, CA, USA) was mixed with 100 μL of eluate in a 2 mL tube and incubated for 1 h at 37 °C;
2. At the end of incubation, 180 μL of AMPure XP by Beckman Coulter (Bray, CA, USA) was added to the eluate, mixed gently by flicking the tube, and drops were then separated by spinning, and the tubes incubated for 5 min;
3. The tubes were placed on a magnetic rack until discoloration of the liquid was observed. Then the supernatant was removed;
4. The precipitate was washed twice with 300 μL of freshly prepared 70% alcohol;
5. Purified DNA was eluted in 100 μL of nuclease-free water by NEB (Ipswich, MA, USA), mixed gently by flicking the tube, and incubated for 5 min, after dissolving the precipitate.

We used a Short Read Eliminator Kit by Circulomics Inc. (Baltimore, MD, USA) to enrich the library with long fragments. The quality of isolated and purified DNA was tested on a Nabi UV/Vis Nano Spectrophotometer.

### 2.3. Sequencing of Genomic DNA

Sequencing was performed with the SQK-LSK109 kit using MinION Mk1C and FLO-MIN106 cell by Oxford Nanopore Technologies (Oxford, UK). The library was prepared according to the protocol "Genomic DNA by Ligation". Only data obtained by nanopore sequencing were used for this study.

### 2.4. Bioinformatic Analysis

#### 2.4.1. Data Filtering

Guppy [28] was used to extract the FASTQ sequences from the 5 fast signals. The resulting reads were filtered via NanoFilt [29] with a minimum read length of 600 bp and quality above 7. NanoFilt was applied to remove 40 bp at each end of the reads. The DM v6.1 assembly of the DM 1-3 516 R44 double monoploid potato genome [22] was used as a reference.

#### 2.4.2. SVs Calling

The filtered reads were aligned to the reference genome using NGMLR [30].

Sequencing depth was estimated in bamCoverage [31] and visualized in IGV [32]. Variant calling was performed with the options described in Table 1 using SVIM (Structural Variant Identification by Mapped Long Reads) [33] and Sniffles [30] algorithms, which use different approaches to search for structural variants. Sniffles uses split-read alignments to search for SVs, while SVIM searches for SVs in each read and then combines them.

**Table 1.** Summary of the quality table of the obtained reads.

| Parameter | SVIM | | Sniffles | |
|---|---|---|---|---|
| | Option | Value | Option | Value |
| Minimum SV length | –min_sv_size | 3 | –l | 3 |
| Maximum SV length | –max_sv_size | 100,000,000 | — | — |
| Minimum reads number for SV determination | –minimum_depth | 20 | –s | 10 |
| Minimum quality | –min_mapq | 40 | –q | 40 |
| Maximum distance to group SVs together | –segment_gap_tolerance | 5 | –d | 5 |

#### 2.4.3. SV-Gene Matching

To search the indels enclosed within genes, we used annotation data based on DM High Confidence Gene Model Set v6.1 [34].

Data were visualized and processed in R using packages ggplot2 [35] vcfR [36], seqinr [37] and VennDiagram [38].

## 3. Results

### 3.1. Alignment of Three Potato Varieties' Genomes against Reference

We obtained approximately 8.5 million reads with an average length of 51 gigabases per sample. After filtering, we retained ca. 7.6 million reads with 44 billion nucleotides in total.

The proportion of reads aligned to the reference genome was 72.3% for the variety Argo, 74.1% for Shah, and 73.8% for Alaska. The whole reference genome was covered at least 40 times. The remainder of the reads belonged to mitochondrial and plastid genomes, as well as indeterminate repetitive multichromosomal regions. The results of sequencing and filtering are shown in Table 2.

**Table 2.** Summary of the quality table of the obtained reads.

| Variety | Number of Reads | Total Reads Length, Gbp | Mean Read Length, bp | Max Read Length, bp | Mean Read Quality | Coverage [1] |
|---------|-----------------|-------------------------|----------------------|---------------------|-------------------|--------------|
| Alaska | 7,009,345 | 42 | 5992 | 138,417 | 22,5 | 42 |
| Argo | 7,916,456 | 47 | 5937 | 142,819 | 21,3 | 46 |
| Shah | 7,841,739 | 44 | 5611 | 119,045 | 20,8 | 44 |

[1] The length of DM v6.1 reference assembly is 740 Mbp.

### 3.2. Finding Structural Variants

We used filtered and aligned reads to investigate structural variants in the genomes of studied varieties.

SVIM and Sniffles require different approaches to filtering. The VCF-file provided by Sniffles does not have a QUAL column, so quality control is available only in the Sniffles option. We selected values of 40 and 20 on the Phred-scaled quality score for Sniffles and SVIM, respectively, as a trade-off between quality and SV numbers. Estimation of sequencing depth also differed for SVIM and Sniffles, where the former estimates depth without considering indels, and the latter estimates the exact read coverage. So, the difference between both SV callers comprised 1.5–2 times. Therefore, we have chosen minimum depths of 20 and 15 for SVIM and Sniffles, respectively, and removed sequences with excessive read depth. Overrepresentation of any SV can indicate an unspecific alignment of the mitochondrial and plastid genomes with the nuclear genome.

The total numbers of SVs detected by SVIM/Sniffles were 34,523/35,761, 57,614/57168, 44,876/44,674 for Alaska, Argo, and Shah, respectively. The sequencing coverage can explain the difference in the number of SVs between varieties (e.g., Argo has the highest coverage and the highest number of SVs). Both algorithms found approximately the same number of SVs. We classified SVs into three groups: short (4 bp–5 kbp), medium (5–100 kbp), and large (over 100 kbp). Short SVs were detected by both methods in approximately equal numbers. However, SVIM was less sensitive to indels larger than 5 kbp. In addition, in comparison with SVIM, Sniffles was more sensitive to duplications, revealed deletions, insertions, and inversions longer than 100 kbp (Figure S1). The total numbers of structural variants are presented in Table 3.

Deletions and insertions are the most common SVs found, while duplications and inversions are the least represented. Large inversions involving vast parts of chromosomes are the most common among large SVs. The sequencing depth was almost equal for the whole length of each chromosome. Nevertheless, the distribution of SVs within the chromosomes was uneven and correlated with regions of euchromatin and heterochromatin (Figures S2 and S3). The SV density was significantly reduced in the central part of the chromosomes as compared to the edges.

**Table 3.** Numbers of structural variants in the genomes of Alaska, Argo, and Shah varieties. Deletions/insertions/duplications/inversions.

| Variety | SVIM | Sniffles |
|---|---|---|
| Short SVs | | |
| Alaska | 17,809/16,686/-/- | 20,472/15,000/32/6 |
| Argo | 30,161/27,393/4/- | 33,934/22,806/50/9 |
| Shah | 22,391/22,433/7/- | 25,708/18,674/32/5 |
| Medium SVs | | |
| Alaska | 28/-/-/- | 207/1/9/8 |
| Argo | 55/-/-/- | 315/-/14/11 |
| Shah | 42/-/-/- | 220/-/9/7 |
| Large SVs | | |
| Alaska | -/-/-/- | 4/-/2/22 |
| Argo | 1/-/-/- | 6/-/2/24 |
| Shah | 3/-/-/- | 3/-/4/12 |

The numbers of indels identified by both SVIM and Sniffles are 16,438, 29,204, and 22,069 for Alaska, Argo, and Shah, respectively (Figure 1). The SVIM–Sniffles indels comprised 24.6% of all found SVs. There were no common duplications or inversions. The largest number of indels was identified for Argo. For this reason, this variety has more SVs similar to those of other species. Only 9.8% of the SVIM–Sniffles indels were common for all three varieties.

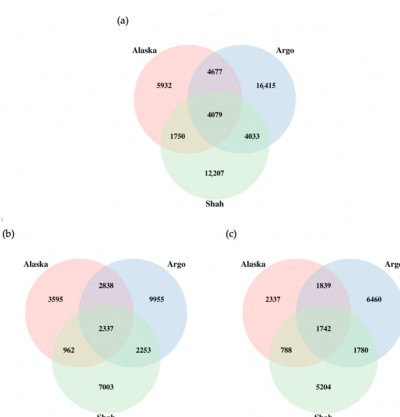

**Figure 1.** The unique and common SVIM–Sniffles structural variations for different potato varieties. (**a**) All indels; (**b**) deletions; (**c**) insertions.

### 3.3. Structural Variants into Coding Sequences

Table 4 shows the results of the SVs and matching potato genes. Almost half of the detected indels up to 5 kbp in length lie within genes. At the same time, short indels with lengths of up to 30 bp have the greatest weight; the proportion of such deletions and insertions is higher than 78% and 81%, respectively.

**Table 4.** Numbers of structural variants in the genome of Alaska, Argo, and Shah varieties. Deletions/insertions/duplications/inversions.

| Variety | SVIM | Sniffles | SVIM–Sniffles |
|---|---|---|---|
| Alaska | 8106/7398/-/- | 9274/7302/9/1 | 4747/3410/-/- |
| Argo | 13,381/12,069/-/- | 15,082/10,886/21/1 | 8236/5884/-/- |
| Shah | 10,070/10,188/3/- | 11,451/8987/12/1 | 6000/4857/-/- |

Approximately one-third of all potato genes carry mutations (9179/8436, 12,821/11,800, and 11,068/9757 in Alaska, Argo, and Shah varieties, respectively, based on SVIM/Sniffles algorithms). The proportions of genes affected by deletions and insertions are 19.8%, 26.7%, and 24.5% in Alaska, Argo, and Shah, respectively.

Most of the indels belong to introns and 3′-UTRs and should not have a visible effect (Figure 2). However, about 1% of such indels are longer than 1500 bp, which leads to the loss or alteration of the 3′-UTR, exons or entire genes and potentially could result in loss of function. In addition, we found 600 deletions and 500 insertions, located either in coding sequences (CDS and exons) or in the 5′-UTR regulatory region per variety.

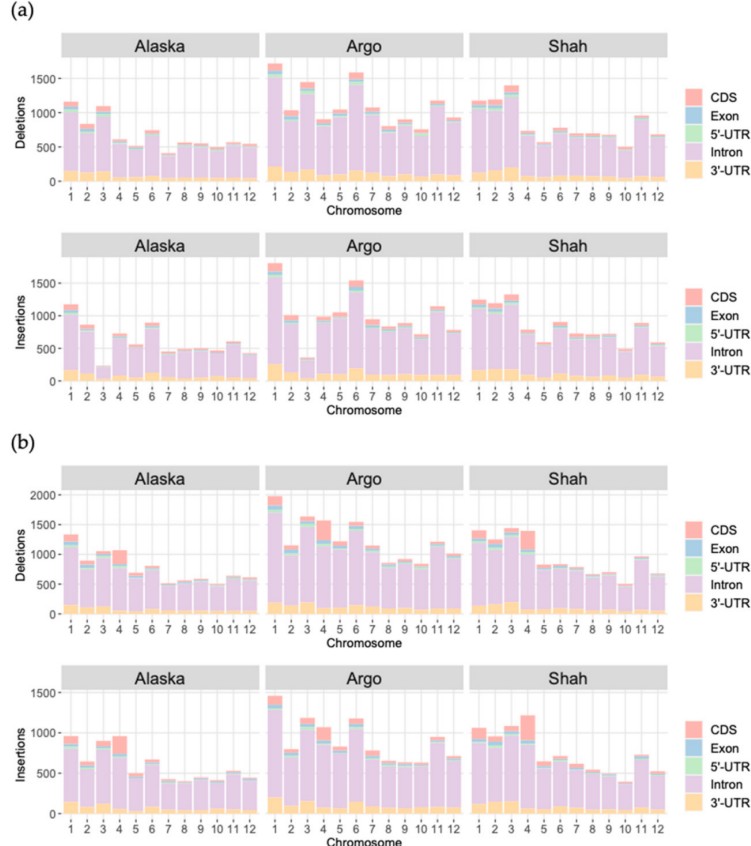

**Figure 2.** The numbers of structural variants identified by (**a**) SVIM and (**b**) Sniffles algorithms.

Large deletions and duplications (2.5 kbp–55 Mbp) identified by Sniffles affect whole genes and their clusters. Indels leading to duplications or deletions of one to two genes are more frequent, but their total contribution to the change in the number of copies is much lower than that of SVs longer than 100 kbp. (Table 5). We identified the encoded proteins or protein families for some deletions and duplications (Table S1). For example, dehydration-responsive protein (protein of drought stress [39]), ubiquitin-related protein (resistance to pathogens, response to abiotic stress [40]), polygalacturonase inhibiting protein (resistance to pathogens [41]), and β-fructofuranosidases (which play a role in the cold-induced sweetening of potato tubers [42]) were duplicated. In contrast, some of the nucleic acid synthesis-related proteins, metabolic enzymes and no apical meristem (NAM) protein (protein of salt and heat stress, resistance to *P. infestans* [43]) were deleted.

**Table 5.** Numbers of genes in the SVs. Deletions/duplications.

| Variety | Number of Genes | | | Number of SVs | | |
|---|---|---|---|---|---|---|
| | **Total** | **>100 kbpSVs** | **<100 kbpSVs** | **Total** | **>100 kbpSVs** | **<100 kbpSVs** |
| Alaska | 2594/170 | 2540/135 | 54/35 | 29/10 | 3/2 | 26/8 |
| Argo | 1498/731 | 1435/680 | 63/51 | 41/17 | 5/2 | 36/15 |
| Shah | 926/1336 | 892/1305 | 34/31 | 26/13 | 3/2 | 23/11 |

## 4. Discussion

In the current study, we sequenced the genomes of three potato varieties selected at the Research Center by Shanina E.P. and Klyukina E.M. (Ural Federal Agrarian Research Center Ural Branch of the Russian Academy of Science, Ekaterinburg, Russia) and we found their structural variants. All the varieties have biotic and abiotic stress tolerance traits, making them applicable in the Urals and regions with similar climatic conditions and soil.

The potato genome shows a high degree of diversity at the SV level. Using the SVIM and Sniffles algorithms, we detected more than 30,000 SVs against DM v6.1 reference for each variety based on a large number of high-quality sequencing reads. The total number of the indels detected by both algorithms was more than 24,000 per variety, with the highest number of SVs in Argo, and the lowest in Alaska. Noteworthy, these differences could be caused by the difference in the sequencing coverage and depth. The coverage and the depth could also affect the number of detected indels.

Previous research did not include SVs shorter than 500 bp [23,25], due to low coverage, meanwhile SVs shorter than 500 bp made up more than 95% of all found SVs, in concordance with recent research using a combination of sequencing techniques [24], and demonstrating that the majority of all SVs can be shorter than 50 bp. On the other hand, there is evidence that SVs larger than 100 kbp are common in the potato genome [23,44], meanwhile, we found around 20 large SVs per variety and, notably, only a minority of SVs belonged to deletions and duplications. There are a few possible reasons for this, including: obtained coverage was insufficient to determine large deletions and duplications; read length was less than necessary to cover an SV's region at alignment; SVIM and Sniffles did not enable accurate detection of SVs in a polyploid genome. Therefore, a higher coverage could improve the detection of large SVs. As previously reported in other studies of the potato genome [23–25], deletions prevail over the other types of SVs in intergenic and genic regions.

The proportions of SVs were almost equal in intergenic and genic-intragenic regions. About 50% of SVs shorter than 5000 bp affected up to one-third of all genes in the genome. However, only about 1000 indels within exons, critical regulatory regions of the transcripts, or entire genes could directly affect expression, e.g., cause frame-shift mutations. This kind of mutation might affect gene expression [45]. The results correspond to early studies of the structural variants in the potato genome [23–25], with the distinction that our work was conducted using nanopore sequencing. About 40 large SVs included over 2000 genes and could lead to changes in the gene copy number and the gene expressions in each variety. We detected deletions in regulatory proteins of replication, methylations, and some metabolic proteins. In the genome of the potato, there are about a hundred NAM-related proteins [42], so deletions of this protein are unlikely to have a significant impact on the adaptability of varieties. We found duplications of the dehydration-responsive protein and some ubiquitin-related proteins in all the varieties, explaining their resistance to drought and other types of abiotic stress. We also detected duplication of polygalacturonase- inhibiting protein in Alaska's genome. We assume this mutation determines the resistance of the variety to late blight. The revealed duplication of β-fructofuranosidase in Argo can affect the storage period.

These results form the basis for further research aimed at detecting more target genes that can determine storage longevity, yield, resistance to abiotic stress, and pathogenic microorganisms related to particular geographic regions.

**Supplementary Materials:** The following are available online at https://www.mdpi.com/article/10.3390/agronomy11091703/s1, Figure S1: Karyotyping of SVs larger than 5 kbp on potato chromosomes, Figure S2: Distribution of deletions and insertions on potato chromosomes (a) identified by Sniffles; (b) identified by SVIM, Figure S3: Sequencing depth and gene distribution per chromosome, Table S1: List of structural variants containing a few genes found in Alaska, Argo and Shah.

**Author Contributions:** Conceptualization, all authors; methodology, all authors; validation, G.A.L.; investigation, G.A.L.; writing—original draft preparation, G.A.L.; writing—review and editing, E.P.S.; visualization, G.A.L.; supervision, E.P.S.; funding acquisition, E.P.S. All authors have read and agreed to the published version of the manuscript.

**Funding:** This study was funded by the Ministry of Science and Higher Education of the Russian Federation under agreement No. 0773-2020-0022, which provides a grant in the form of subsidies from the Federal budget of the Russian Federation. The grant was provided within the framework of the State Target "Development of competitive, high-yielding, world-class varieties of grain, leguminous, forage, fruit and berry crops and potatoes based on promising genetic resources that are resistant to bio- and abiotic factors".

**Institutional Review Board Statement:** Not applicable.

**Informed Consent Statement:** Not applicable.

**Acknowledgments:** The authors are grateful to Bogatova P.S., Mikryukov V.S., Dulya O.V. and Modorov M.V. for their help and valuable comments on the manuscript.

**Conflicts of Interest:** The authors declare no conflict of interest.

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
