# Peer review of "Structural Variations in the Genome of Potato Varieties of the Ural Selection"

_agronomy, doi:10.3390/agronomy11091703_

Round 1

Reviewer 1 Report

The authors have identified over 24,000 structural variants ranging from 30 to 10,000 bp in length in the three genomes of potato with the goal to identify molecular markers for further genetic selection. After I have carefully reviewed the manuscript, I strongly believe that the manuscript could benefit significantly from the following potential improvement:

  1. The Abstract needs to be largely improved because the main results are oversimplified, and the significance of this study is missing. The word “Russian” should be removed because it is redundant since “in the world” immediately prior to it was already indicated. Furthermore, one statements saying that “No structural variants larger than 30 bp in coding regions of the genome of the studied varieties were found.” was not found in the Results. This is confusing.
  2. In the Introduction, I would like to see some explanations of the biological significance of structural variants in the genome of potato.
  3. Lines 31-33, the latin names should be italicized, and the first use of Solanum should not be abbreviated.
  4. Line 95, “Guppy” needs a reference
  5. Table 2, I would suggest that the authors combine the information of Small SVs, Medium SVs, and Large SVs. For example, the “Deletions of Argo” of Small SVs, Medium SVs, and Large SVs could be presented as 20,331/245/0 together in one cell This could make the information and the table less redundant.
  6. The results presented in Table 2 and Figure 1 need more detailed explanation, what is the difference between SVIM and Sniffles?
  7. The major problem that I have seen in this manuscript is that the Discussion contained simply the briefly repeated results but lacking any serious discussion of the results, in particular the results presented in Figures 1 and 2 and Table 2 in relation to related studies.

Also, I have seen quite some grammatical and editorial errors, including some incorrect or informal use of words. I would highly recommend that the authors should carefully proofread the entire manuscript if a revision is required by the Editor.

Author Response

Thanks for your review.

We rewrote the paper.

  1. We corrected and supplemented the abstract in accordance with the conclusions.
  2. We supplemented the introduction, tried to make it more coherent and understandable.
  3. Corrected Latin names.
  4. Add the reference
  5. Thank you for your advice, we rewrote the tabels.
  6. We tried to explain main difference between SVIM and Sniffles in 2. Materials and Methods.
  7. Rewrote conclusions.
  8. We tried to correct grammatical errors

Reviewer 2 Report

The authors need to make some corrections, as mentioned below.

English language corrections are required from a Native speaker
Add more information in the abstract.
Restructure Introduction include a solid hypothesis.

Replace old references with recent ones. 
Add more information in the Introduction.
Material and methods are not clear; please elaborate
The figures are poorly presented and difficult to interpret.
Elaborate on the discussion section.

Author Response

Thanks for your review.

We rewrote the paper.

  1. We tried to correct grammatical errors.
  2. We corrected and supplemented the abstract in accordance with the conclusions.
  3. We supplemented the introduction, tried to make it more coherent and understandable.
  4. We supplemented the Materials and Methods. We tried to explain main difference between SVIM and Sniffles.
  5. We have replaced figures and tables more readable and informative.
  6. Rewrote conclusions.

Round 2

Reviewer 1 Report

I appreciate very much the efforts that the authors have devoted to improving their manuscript. However, I still think that the Discussion needs a lot of work to do. Again, the Discussion only contains the repeated results without any meaningful discussion of the biological importance of the results revealed in this study.

Author Response

Thank you for your review.

We rewrote Discussion, we added the comparison with previous studies and we tried to explain our results better than last time.

Reviewer 2 Report

Thanks for the correction, still Discussion and Conclusions are strangely combined by the authors and even after doing so, the section is so small like conclusions. I request the authors to restructure this section and add more information appropriately.

Author Response

(The authors gave the same response as above.)
